# High-Figure-of-Merit Biosensing and Enhanced Excitonic Absorption in an MoS_2_-Integrated Dielectric Metasurface

**DOI:** 10.3390/mi14020370

**Published:** 2023-02-01

**Authors:** Hodjat Hajian, Ivan D. Rukhlenko, A. Louise Bradley, Ekmel Ozbay

**Affiliations:** 1School of Physics, CRANN and AMBER, Trinity College Dublin, D02 PN40 Dublin, Ireland; 2Institute of Photonics and Optical Science (IPOS), School of Physics, The University of Sydney, Camperdown, NSW 2006, Australia; 3Information Optical Technologies Centre, ITMO University, Saint Petersburg 197101, Russia; 4IPIC, Tyndall National Institute, T12 R5CP Cork, Ireland; 5Nanotechnology Research Center (NANOTAM), Institute of Materials Science and Nanotechnology (UNAM), Department of Physics, Department of Electrical and Electronics Engineering, Bilkent University, Ankara 06800, Turkey

**Keywords:** MoS_2_, dielectric metasurfaces, broken in-plane inversion symmetry, quasi-bound states in the continuum, biosensing, enhanced excitonic absorption

## Abstract

Among the transitional metal dichalcogenides (TMDCs), molybdenum disulfide (MoS_2_) is considered an outstanding candidate for biosensing applications due to its high absorptivity and amenability to ionic current measurements. Dielectric metasurfaces have also emerged as a powerful platform for novel optical biosensing due to their low optical losses and strong near-field enhancements. Once functionalized with TMDCs, dielectric metasurfaces can also provide strong photon–exciton interactions. Here, we theoretically integrated a single layer of MoS_2_ into a CMOS-compatible asymmetric dielectric metasurface composed of TiO_2_ meta-atoms with a broken in-plane inversion symmetry on an SiO_2_ substrate. We numerically show that the designed MoS_2_-integrated metasurface can function as a high-figure-of-merit ($\mathrm{FoM}=137.5\;{\mathrm{RIU}}^{-1}$) van der Waals-based biosensor due to the support of quasi-bound states in the continuum. Moreover, owing to the critical coupling of the magnetic dipole resonances of the metasurface and the A exciton of the single layer of MoS_2_, one can achieve a 55% enhanced excitonic absorption by this two-port system. Therefore, the proposed design can function as an effective biosensor and is also practical for enhanced excitonic absorption and emission applications.

## 1. Introduction

Metasurfaces—two-dimensional (2D) metamaterials composed of subwavelength resonators (meta-atoms)—can be employed to tailor the amplitude, phase, and polarization of incident electromagnetic waves. Based on these outstanding characteristics, they have been used in many applications, such as polarization conversion, holography, optical vortex generation, lensing, and beam splitting [1].

Dielectric metasurfaces (DMs) have been recently employed for flat optics wavefront manipulation due to their low resonance-induced heating as compared to their plasmonic counterparts and CMOS-compatible fabrication processes [2,3,4]. Remarkably, asymmetric DMs composed of meta-atoms with broken in-plane inversion symmetry can support high-Q resonances originating from the symmetry-protected quasi-bound states in the continuum (q-BIC) [5,6,7,8]. A true BIC is a mathematical object with an infinite value of the Q factor and a vanishing resonance width. BICs are inaccessible by external excitations, but they can be realized in practice as q-BICs when both the Q factor and resonance width become finite [6,8]. In other words, as they are the radiative continuum, the BIC-associated modes could potentially be accessible from the free space when they become q-BICs with a weak yet perceptible interaction with the continuum. While they can be supported by metallic metasurfaces [9], the high-Q (q)-BICs of DMs can also boost the electric field enhancement inside the structure with low losses and be employed for applications such as lasing [7], enhanced nonlinear harmonic generation [8], and light–matter interaction in van der Waals (vdW)-integrated DMs [10,11,12]. Label-free biosensing is another useful application of DMs. In the following, we first explain the different types of label-free biosensors, and then review optical biosensors based on metasurfaces and DMs.

Optical biosensors are widely used for biomedical research, healthcare, and environmental monitoring. Compared with standard optical sensors based on fluorescence measurement, label-free nanophotonic sensors do not need to label biomolecules, which reduces the effort involved in sample preparation and enables rapid analysis in real time [13,14,15,16,17]. The performance of a sensor can be characterized by the detection limit, which is inversely proportional to the product of the quality factor (Q) of the resonance and the refractive index sensitivity ($S=\frac{\Delta \lambda_{res}}{\Delta n_{s}}\;\left[\frac{\mathrm{nm}}{\mathrm{RIU}}\right]$) [14]. Consequently, a higher Q×S yields lower detection limits. A figure-of-merit ($\mathrm{FoM}=\frac{S}{\mathrm{FWHM}}\;\left[{\mathrm{RIU}}^{-1}\right]$) is an alternative widely adopted metric for the intrinsic resolving power of metasurface-based optical biosensors. Here, $\Delta \lambda_{res}$ and FWHM stand, respectively, for the peak wavelength shift with the change in the refractive index $\Delta n_{s}$ of the sensing medium and the full width at half maximum/minimum resonant peak/dip.

Commercially available biosensors based on surface plasmon resonance (SPR) are the most notable and widely studied category of refractive index sensors [18]. In these conventional SPR-based sensors, gold or silver is deposited directly on the base of a prism to separate the sensing medium from the prism. Gold is regarded as the most suitable material due to its good resistance to oxidation and corrosion, great chemical stability, and superior optical performance. However, the ability of gold to bind biomolecules is poor, which limits the sensitivity of conventional prism-Au-based SPR biosensors. Furthermore, high Ohmic losses broaden the supported resonances, resulting in low FoMs. To overcome this issue, plasmonic/phononic metasurfaces based on van der Waals (vdW) materials—namely, graphene [19] and hexagonal boron nitride [20]—and metals [21,22,23,24,25,26,27,28] have been proposed. However, the intrinsic losses can still reduce the performance of these sensors.

To remove the negative effect of Ohmic losses, specifically for operation in the visible and near-infrared ranges, ring resonators [29], photonic crystals [30], and dielectric metasurfaces have been employed in new generations of optical biosensors. Devices based on ring resonators and photonic crystals, with in-plane excitations, have been explored for refractive index sensing and to build compact and high-performance on-chip integrated sensors. However, a delicate alignment is needed to couple light from a fiber into an on-chip waveguide in such systems. It has been demonstrated that the coupling of the out-of-plane free-space incident light with guided resonances [31], BICs [32,33], and Mie resonances [34,35,36] supported by PhCs and DMs [37] with in-plane symmetry can be an alternative effective strategy for high-FoM label-free biosensing.

High-Q q-BICs supported by CMOS-compatible asymmetric DMs, composed of arrays of meta-units with broken in-plane inversion symmetry, have been recently employed for high-FoM label-free biosensing and hyperspectral imaging [38,39,40,41]. Recent studies have shown that an ultrasensitive label-free analytical platform for biosensing can be achieved by combining such dielectric metasurfaces and hyperspectral imaging. Using this technique, spatially resolved spectra from millions of image pixels are acquired, and smart data-processing tools are employed to extract high-throughput digital sensing information at the unprecedented level of less than three molecules per ${\mu m}^{2}$ [38,39].

In addition to using low-loss or lossless meta-atoms, employing materials that are compatible with biomolecular recognition elements (BREs), such as graphene [42], black phosphorus [43], transitional metal dichalcogenides (TMDCs) [44], and hybrid graphene-TMDC heterostructures [45], is another strategy to enhance the performance of biosensors. Monolayers of TMDCs are natural direct bandgap 2D semiconductors exhibiting strong excitonic responses even at room temperature due to the support of excitons with large binding energies and oscillator strengths [46,47]. Moreover, TMDCs are uniquely compatible with diverse substrates with minimum lattice matching restrictions [48,49]. In the presence of these materials, the adsorption of biomolecules to the surface of the sensor is enhanced due to their high surface-to-volume ratio and the vdW forces on the surface. This enhances the sensor performance [44]. Among TMDCs, molybdenum disulfide (MoS_2_) is considered an appropriate candidate for biosensing. Being composed of an alternative arrangement of the pore atoms (Mo/S) with a hydrophobic–hydrophilic–hydrophobic architecture makes MoS_2_ amenable for ionic current measurements with lower noise and a high affinity for protein-surface adsorption and, therefore, practical for biosensing applications [50,51].

In this study, we first examined the transmission and reflection spectra of a CMOS-compatible asymmetric DM composed of TiO_2_ meta-atoms with a broken in-plane inversion symmetry. Through the investigation of the asymmetry parameter of the metasurface, as well as its electric and magnetic field profiles, we showed that the q-BIC and magnetic dipole (MD) modes are responsible for the resonant responses of the metasurface. Finally, by considering the above-mentioned features of MoS_2_ for sensing purposes, we integrated a single layer (1L) of MoS_2_ into an appropriately designed DM. A 55% enhancement in the excitonic absorption was achieved in the 1L-MoS_2_-integrated dielectric metasurface (MoS_2_-DM) due to the critical coupling of the MD of the dielectric metasurface and the A exciton of the 1L MoS_2_. Moreover, owing to the support of the q-BIC by the DM, high values of sensitivity ($S=222\;\frac{\mathrm{nm}}{\mathrm{RIU}}$) and a high-figure-of-merit ($\mathrm{FoM}=137.5\;{\mathrm{RIU}}^{-1}$) were obtained at $\lambda_{q-\mathrm{BIC}}$, proving the biosensing capability of the designed vdW-based biosensor.

## 2. Dielectric Metasurface

We started our analysis by investigating the transmission and reflection spectra of a bare dielectric metasurface and analyzing the electric and magnetic field profiles of its resonant modes. As shown in Figure 1a, the bare metasurface consists of TiO_2_ nanoresonators on an SiO_2_ substrate, which can be fabricated with the CMOS-compatible processes [4,38,39]. The unit cell of the DM is composed of two elliptical TiO_2_ nanoresonators separated by gap g and tilted by angle θ, as shown in Figure 1b. The length, width, and height of the resonators are l, w, and h. We performed finite-difference time domain (FDTD) simulations [52] using periodic boundary conditions in the X and Y directions and a perfectly matched layer in the Z direction. The metasurface was excited by an X-polarized plane wave, and two symmetrically located 2D monitors were used to obtain the transmission and reflection spectra.

Figure 2a,b show how the asymmetry parameter affected the transmission and reflection spectra of the bare DM excited by normally incident x-polarized light. One can observe that the bare DM supported two resonances. The upper dispersion branches (the dark and bright regions in Figure 2a,b, respectively) were almost flat and came from the MD resonances, as will be discussed below. The lower branches were considerably more dispersive with respect to the asymmetry parameter, which indicated the BIC origin of theses resonances. In other words, the sharp spectral resonances of the lower branch are controlled by θ such that the Q-factor grows to infinity when the tilting angle vanishes. This is a well-known characteristic of q-BIC resonances [6].

In support of this statement, Figure 2c shows the FWHM of the q-BICs as a function of the asymmetry parameter. Figure 2d shows the transmission (solid blue) and reflection (solid red) spectra for θ=17.5°, where a balance between the strength of the resonance and the corresponding FWHM was achieved. For this relatively small value of the tilting angle, two high-Q resonances at λ=628 nm (the q-BIC mode) and 658.8 nm (the MD mode) existed. The FWHMs of these resonances were 3.6 nm (Q=165) and 2.7 nm (Q=240), respectively. It is also noteworthy that both resonances could be spectrally tuned by changing other geometrical parameters, as indicated in Figure 1b.

To gain a deeper insight into the interaction of the incident plane wave with the bare DM, we analyzed the distributions of the electric and magnetic fields across the unit cell, as shown in Figure 3 and Figure 4. As seen from the top views of $\left|E\right|$ and $\left|H\right|$ at the surface of the DM in Figure 3a,b, the electric and magnetic fields were mostly localized outside of the nanoresonators and had high levels of enhancement. Therefore, owing to their spatial overlap with the surface-confined electromagnetic fields, the DM could be extremely responsive to the local refractive index changes induced by the individual biomolecules covering the surface. This characteristic makes the DM operating at the q-BIC resonance very appealing for sensing applications [6,38,39].

Figure 3c shows that the electric field was captured inside the nanoresonator around z=0.1 μm and flowed in the y direction, while the magnetic field circulated clockwise and counterclockwise inside the left and right nanoresonators, respectively (Figure 3d). This feature indicates the electric dipole nature of the supported q-BIC.

For the resonant mode at 658.8 nm, the top-view electric field was mostly localized in the proximity of the edges of the resonators (Figure 4a), while the magnetic field was mostly localized in the gap between them (Figure 4b). Consequently, this resonant mode was less sensitive to changes in the refractive index of the covering medium and less suitable than the q-BIC resonance for sensing purposes.

The side-view profiles showed that the fields were captured inside the resonators around z=0.1 μm with circular and x-oriented directions of the electric (Figure 4c) and magnetic (Figure 4d) fields, respectively, which indicated the MD nature of the resonance. When Figure 3 and Figure 4 are compared, it is observed that the enhancement of the electric field at the MD resonance was almost twice the enhancement at the q-BIC resonance. This was a unique feature of our design that led to the critical coupling of the DM with the A exciton of the 1L MoS_2_ at the MD resonance, and to the weak interaction of the DM with the 1L MoS_2_ at the q-BIC resonance.

## 3. 1L-MoS_2_-Integrated Dielectric Metasurface

Various types of TMDCs have recently been integrated into DMs to obtain photon–exciton interaction in either strong or weak coupling regimes (see e.g., Refs. [11,12]). Among TMDCs, MoS_2_ shows outstanding characteristics for biosensing applications, including high adsorptivity [44] and amenability to ionic current measurements [50,51]. Consequently, as described below, we integrated 1L MoS_2_ into the DM (Figure 5a) for the purpose of enhanced excitonic absorption via the critical coupling of exciton–MD resonance and high-FoM vdW biosensing via the quasi-BIC resonance.

The real (solid blue) and imaginary (solid red) parts of the permittivity of the 1L MoS_2_ are shown in Figure 5a. The two peaks in the imaginary part of the permittivity, at 665 nm and 619 nm, correspond to the excitonic features associated with the interband transitions at the K and K’ points of the Brillouin zone. The two excitonic features, denoted by A and B, were mainly attributed to the splitting of the valence band by the spin–orbit coupling [46]. Note that the 1L MoS_2_ was modeled as a 2D sheet of surface conductivity, as follows [44]:(1)$$\sigma_{{\mathrm{MoS}}_{2}}=-i\epsilon_{0}\omega t_{{\mathrm{MoS}}_{2}}\left(\epsilon_{{\mathrm{MoS}}_{2}}-1\right)$$
where $\epsilon_{0}$ is the vacuum permittivity and $t_{{\mathrm{MoS}}_{2}}=0.615\;\mathrm{nm}$. In agreement with Ref. [46], the absorption spectrum of the 1L MoS_2_ is plotted in Figure 5c. It was observed that the A and B excitons were responsible for the absorption resonances and that a maximum absorption of about 6.5% could be achieved with a single layer of MoS_2_. Moreover, the excitonic absorption strength at 664 nm was slightly lower than at 619 nm.

Maximizing absorption at the A exciton’s wavelength is beneficial for enhancing photon–exciton interactions and emission applications [12]. We optimized the DM to bring the A exciton absorption up to 55%, which is a considerable absorption enhancement for a 1L vdW-based two-port system. This absorption resonance, obtained by the critical coupling of the MD resonance and the A exciton, is referred to as MD-A resonance. On the other hand, the absorption at the lower wavelength adjacent to the B exciton resonance was kept as low as 36% to avoid a considerable increase in the FWHM of the q-BIC resonance. These features make MoS_2_-DM practical for enhanced photon–exciton interaction (i.e., enhanced excitonic absorption, see Section 3.1) and high-FoM biosensing (see Section 3.2) applications.

### 3.1. Enhanced Excitonic Absorption

The transmission, reflection, and absorption spectra of MoS_2_-DM as functions of λ and θ are shown in Figure 5a–c, respectively. The dark (bright) regions in Figure 5a (Figure 5b,c) represent the dispersion of the MD-A (upper branches) and q-BIC (lower branch) resonances. The transmission and reflection spectra for this case resembled those of the bare DM shown in Figure 2a,b. However, owing to the presence of the MoS_2_ layer, the structure exhibited absorption at both the q-BIC and MD-A resonances. As Figure 6c illustrates, by properly choosing the value of the asymmetry parameter ($\theta =17.5^{\circ }$), it was possible to obtain the maximum absorption at the MD-A resonance while keeping both the absorption and the FWHM of the transmission spectrum low at the q-BIC resonance. Because of this geometrical control over the light absorption, the increase in the FWHM of the q-BIC resonance in the transmission spectrum was kept as low as possible. A comparison of Figure 6d and Figure 2c shows that the FWHM of the q-BIC resonance of the MoS_2_-DM was approximately 2 nm larger than in the case of the bare DM. Therefore, MoS_2_-DM is highly applicable for biosensing applications, as will be discussed in Section 3.2.

Figure 6e shows the transmission, reflection, and absorption spectra of the MoS_2_-DM for $\theta =17.5^{\circ }$. The presence of 1L MoS_2_ resulted in a 40% reduction in the reflection due to the 36% light absorption at $\lambda_{q-\mathrm{BIC}}$ compared to the case of Figure 2d. However, the presence of 1L MoS_2_ did not distort the dip in the transmission spectrum at the q-BIC resonance, with $T_{min}=0.06$ and FWHM=5.6 nm. This feature ensures the attractiveness of the structure for biosensing applications where the sensitivity is measured based on the shifts of the dips in the transmitted signal (see Figure 7).

Due to the coupling of the MD resonance of the dielectric metasurface and the 1L MoS_2_ A exciton, 55% light absorption was achieved at $\omega_{0}=\frac{2\pi c}{\lambda_{MD-A}}$, where $\lambda_{\mathrm{MD}-A}=662\;\mathrm{nm}$ (see the solid black curve in Figure 6e). Further investigations proved that this resonance could be shifted to the A exciton wavelength for h=333 nm. Using the coupled mode theory (CMT), the absorption of this two-port system can be analytically approximated by the following expression [10,53]:(2)$$A=\frac{2\gamma \delta }{{\left(\omega -\omega_{0}\right)}^{2}+{\left(\gamma +\delta \right)}^{2}}$$
where γ is the radiative rate of the MD resonance, and δ is the dissipative loss rate of the A exciton. The result for γ=3.85 meV and δ=3.81 meV was plotted as a dashed red curve in Figure 6f. The good agreement between the solid black and dashed red curves showed that the enhanced MD-A excitonic absorption occurred as a result of satisfying the critical coupling condition γ≅δ.

### 3.2. High-Figure-of-Merit Biosensing

Finally, we examined the biosensing functionality of the bare DM and the MoS_2_-DM. Figure 7a schematically illustrates the adsorption of target biomolecules and detecting antibodies on the surface of the bare DM. An illustration of the enhanced adsorption of the biomolecules by the DM in the presence of an MoS_2_ layer is shown in Figure 7d. It was expected that this adsorption enhancement would lead to a pronounced change in the bulk refractive index of the sensing/covering medium. This, in turn, would be expected to yield higher values of sensitivity ($S=\frac{\Delta \lambda_{res}}{\Delta n_{s}}\;\left[\frac{\mathrm{nm}}{\mathrm{RIU}}\right]$) and FoM ($\frac{S}{\mathrm{FWHM}}\;\left[{\mathrm{RIU}}^{-1}\right]$) for the MoS_2_-DM case. However, the numerical results in Figure 7 assumed that the refractive index of the sensing medium, $n_{s}$, was identically modified for the bare DM and MoS_2_-DM structures to allow for a direct comparison.

**Figure 7 micromachines-14-00370-f007:**
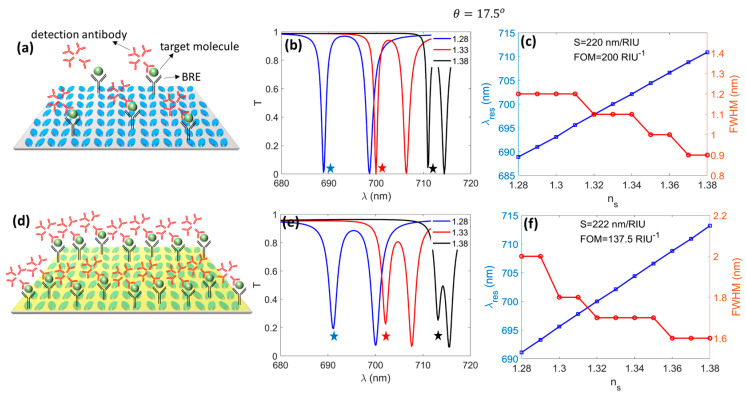
Sensing characteristics of DM without (**a**–**c**) and with (**d**–**f**) 1L MoS_2_: (**a**,**d**) adsorption of biomolecules to the surface of DM; (**b**,**e**) transmission spectra for three refractive indices of the background; (**c**,**f**) wavelength and FWHM of the q-BIC resonance as functions of the background refractive index. The stars in (**b**,**e**) highlight the q-BIC resonances.

Figure 7b shows the resonant transmission spectra of the bare DM for different refractive indices of the sensing medium. It was observed that the change in the refractive index led to noticeable red shifts in the bare DM’s q-BIC resonances (highlighted with stars) while prominently decreasing their FWHMs compared to Figure 2c, as shown in Figure 7c. The obtained values of S and FoM of $220\;\frac{\mathrm{nm}}{\mathrm{RIU}}$ and $200\;{\mathrm{RIU}}^{-1}$, respectively, were noticeably higher than previous reports [38,39]. The integration of 1L MoS_2_ into the DM was seen to result in larger red shifts in the q-BIC resonances and increased FWHMs for the resonances, as was expected due to the introduction of additional losses into the structure. According to Figure 7e, the presence of 1L MoS_2_ increased the sensitivity up to $222\;\frac{\mathrm{nm}}{\mathrm{RIU}}$, while the larger FWHM reduced the FoM to $137.5\;{\mathrm{RIU}}^{-1}$. While lower than the bare DM case, this FoM was sufficiently high for biosensing purposes. Considering that the presence of 1L MoS_2_ manifested in the adsorption enhancement of biomolecules and more effective ionic current measurements, our results suggest that MoS_2_-DM is a practical and effective candidate for high-FoM vdW-based biosensing.

## 4. Conclusions

In summary, we designed a 1L-MoS_2_-integrated asymmetric dielectric metasurface that can function as an enhanced excitonic absorber and a high-FoM vdW-based biosensor. The metasurface is composed of TiO_2_ meta-atoms with broken in-plane inversion symmetry on a SiO_2_ substrate and can be realized with CMOS-compatible processes. Our numerical results showed that the high absorptivity, the amenability to ionic current measurements of MoS_2_, and the presence of high-Q q-BIC resonances rendered the designed metasurface practical for high-FoM ($137.5\;{\mathrm{RIU}}^{-1}$) vdW-based biosensing. It was also shown that the critical coupling of the resonant magnetic dipole of the metasurface and the A exciton of MoS_2_ allowed a 55% enhancement in excitonic absorption. This feature also makes the two-port system practical for enhanced absorption and, thereby, emission applications.

## Figures and Tables

**Figure 1 micromachines-14-00370-f001:**
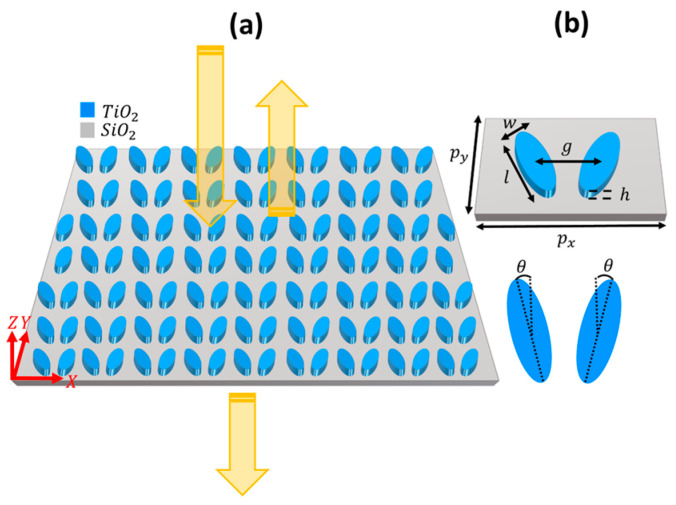
(**a**) Top view of the dielectric metasurface and (**b**) its unit cell with two elliptical dielectric nanoresonators positioned symmetrically with respect to the vertical axis; θ is the asymmetry parameter with respect to the horizontal axis. The arrows indicate the normally incident, reflected, and transmitted x-polarized light. It is assumed that w=95 nm, l=284 nm, h=325 nm, g=266 nm, $p_{x}=422\;\mathrm{nm}$, and $p_{y}=310\;\mathrm{nm}$. The refractive index of SiO_2_ is 1.45.

**Figure 2 micromachines-14-00370-f002:**
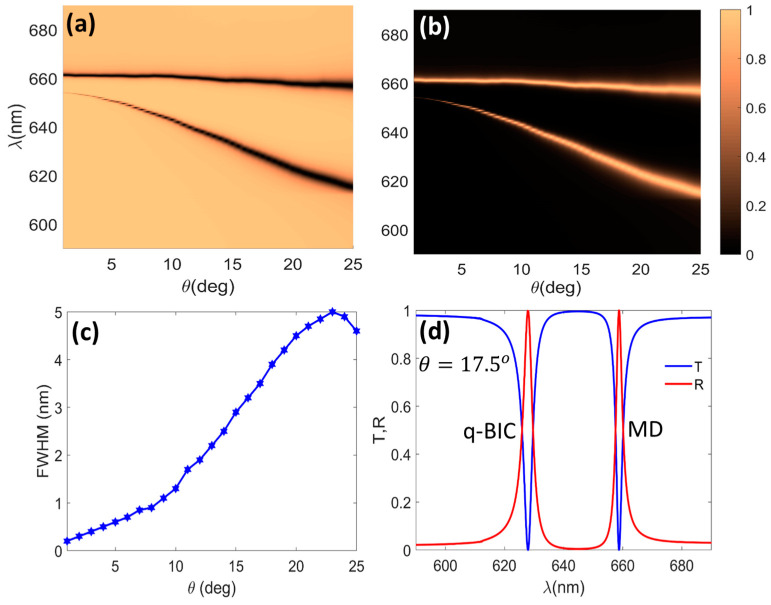
Simulated transmission (**a**) and reflection (**b**) spectra of bare dielectric metasurface vs. the asymmetry parameter θ; (**c**) FWHM of the q-BIC (**lower**) dispersion branch in (**a**) vs. θ. (**d**) Transmission (solid blue) and reflection (solid red) spectra of the DM at $\theta =17.5^{{}^{\circ}}$, for which the q-BIC and MD resonances are located at $\lambda_{q-\mathrm{BIC}}=628\;\mathrm{nm}$ and $\lambda_{MD}=658.8\;\mathrm{nm}$, respectively.

**Figure 3 micromachines-14-00370-f003:**
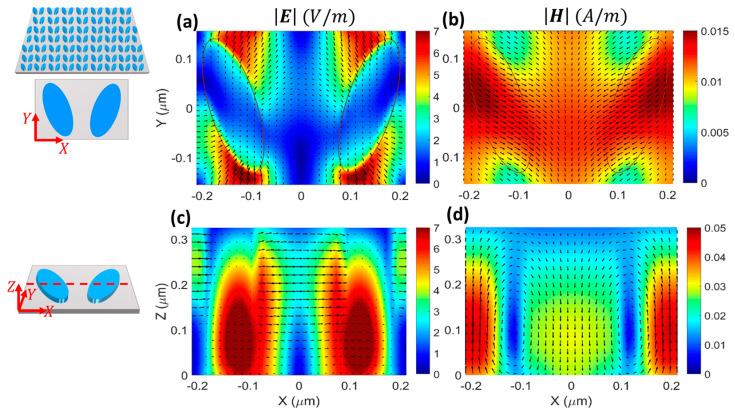
(**a**,**b**) Top view and (**c**,**d**) side view of electric (**a**,**c**) and magnetic (**b**,**d**) fields across the unit cell at the q-BIC resonance $\lambda_{q-\mathrm{BIC}}=628\;\mathrm{nm}$ (Figure 2d). The top view shows the fields on the surface of the nanoresonators (z=h), and the side view represents the fields at the intersection of the 2D monitor, as shown by the red dashed line in the schematic, and the nanoresonators. The overlaid arrows in panels (**a**,**c**) [(**b**,**d**)] indicate $\left|E\right|$ [$\left|H\right|$], and the resonators are located at z=h/2.

**Figure 4 micromachines-14-00370-f004:**
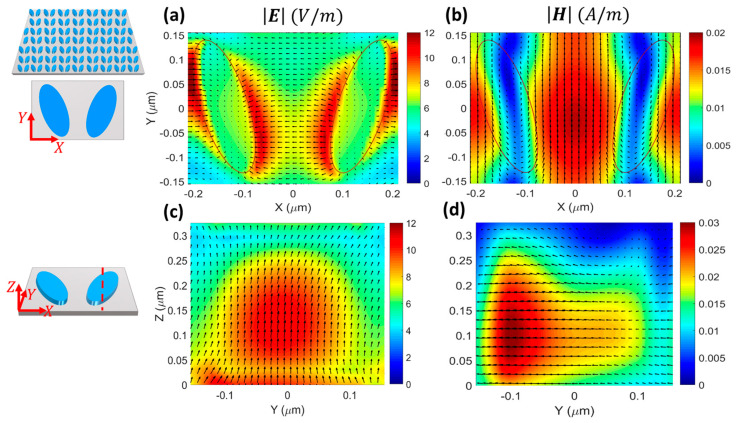
Similar to Figure 3 but for the MD resonance at λ=658.8 nm.

**Figure 5 micromachines-14-00370-f005:**
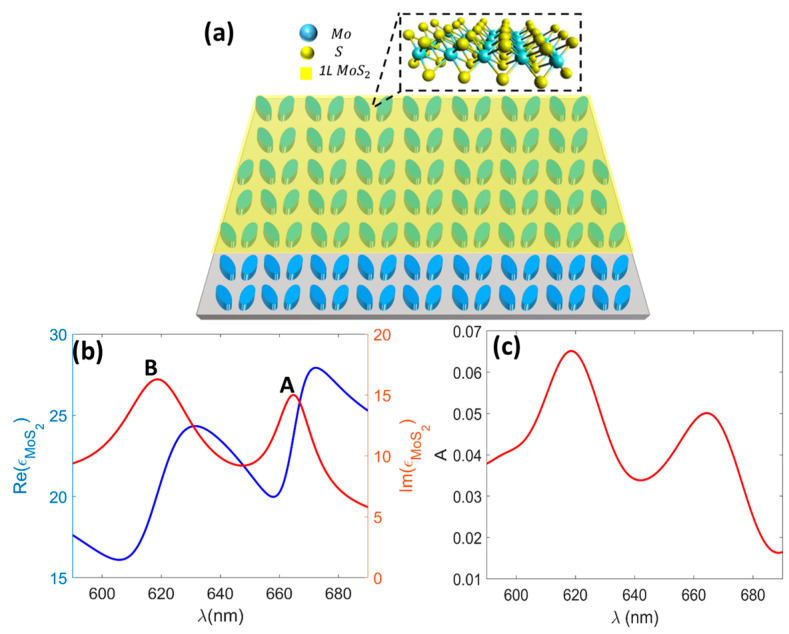
(**a**) Schematic of 1L-MoS_2_-integrated dielectric metasurface, MoS_2_-DM; (**b**) real and imaginary parts of permittivity of the 1L MoS_2_; and (**c**) absorption spectrum of the 1L MoS_2_ on SiO_2_ substrate. Labels A and B indicate excitons with $\lambda_{A}=664\;\mathrm{nm}$ and $\lambda_{B}=619\;\mathrm{nm},$ respectively [46].

**Figure 6 micromachines-14-00370-f006:**
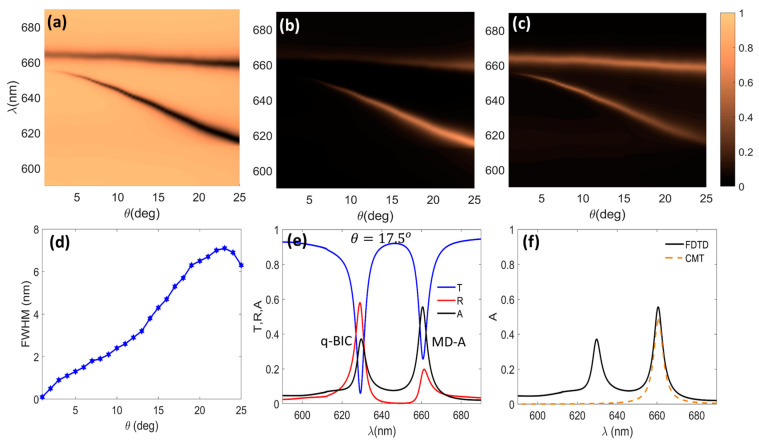
(**a**) Transmission, (**b**) reflection, and (**c**) absorption spectra of 1L-MoS_2_-integrated metasurface; (**d**) FWHM of q-BIC resonances; (**e**) transmission (solid blue), reflection (solid red), and absorption (solid black) spectra of MoS_2_-DM for $\theta =17.5^{\circ }$, $\lambda_{q-\mathrm{BIC}}=629\;\mathrm{nm}$, and $\lambda_{\mathrm{MD}-A}=662\;\mathrm{nm}$; and (**f**) absorption of model two-port system. The upper (**lower**) branches in (**a**–**c**) show the dispersion of the MD-A (q-BIC) resonances.

## Data Availability

The data supporting the findings of this study are available from the corresponding authors upon reasonable request.

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
