# Peer review of "High-Figure-of-Merit Biosensing and Enhanced Excitonic Absorption in an MoS2-Integrated Dielectric Metasurface"

_micromachines, 2023, doi:10.3390/mi14020370_

Round 1
Reviewer 1 Report
The presented manuscript investigated the biosensing performance of a MoS2-integrated dielectric metasurface device. This topic is interesting and timely. However, the manuscript, in its current form, has many flaws, which suggests it cannot be recommended for publication. It can be further considered after a satisfactory major revision.
Major concerns:
1) It is not quite clear which part of the manuscript is based on simulation and which is from experiments. It is required to highlight whether this work is by simulation only or not. This should be mentioned in the title for the reader to understand.
2) Simulating one layer of MoS2 is a challenging task. It is required to further explicitly explain how the simulation was carried out. E. g., what are the physics model and parameters in simulation? Is this setting verified/validated by experiments?
3) If there are any experimental results about the MoS2 layer, pls add at least a SEM image of the device, a Raman for characterizing the MoS2, and experimental setups for measurements.
4) It is a must to benchmark the proposed structure with the state-of-art results to demonstrate the so-claimed “superiority”. Fig. 7 is not enough because it is still unknown how the MoS2-deposited metasurface performs among all counterparts.
Minor concerns:
1) The manuscript should be further proofread.
2) The original definition of FOM is not mentioned. Add ref.
3) Fig. 3&4, add X/Y/z axis in the device structure.
4) Mention how to transfer or deposit monolayer MoS2 on the metasurface.
5) Mention the properties of the SiO2 substrate.
6) Ref are not enough. More papers on recent progress from MDPI journals should be referred to.
Author Response
The Response Letter is attached herewith. Please see the attachment.

Reviewer 2 Report
I suggest the publication of this manuscript.
Author Response

(The authors gave the same response as above.)

Reviewer 3 Report
In the manuscript entitled “High figure of merit biosensing and enhanced excitonic absorption in an MoS2-integrated dielectric metasurface”, H. Hajian et al. have integrated MoS2 single layer to a CMOS-compatible asymmetric dielectric metasurface to be used not only as biosensor but also for emission applications.
First of all, the title, in my opinion, should be re-written since it is poor of significance; I suggest, for example, “MoS2 single layer integrated dielectric metasurface for biosensing and emission applications”.
In the entire manuscript, the English style should be improved.
The abstract is too long, please, eliminate the first paragraph and start from “Here, we integrate…”.
In the introduction, the authors should add more references.
In the introduction, the authors should briefly describe the metasurfaces, specifically the dielectric metasurfaces, reporting some examples and references.
The authors should report separately and not in the text the mathematical equations.
In the introduction, the authors should briefly describe the label free biosensors, also reporting some examples, and explaining how the dielectric metasurfaces could be used for these applications.
In the introduction, the authors briefly report about SPR-based biosensors, but it is not clear how these device work; please, discuss, comparing them to the studied ones, evaluating the advantages and the disadvantages.
The authors should add the experimental section in which they should report details on used materials, experimental conditions, analyses instrumentations and procedures, and so on…
How is the metasurface obtained? Discuss.
How are the transmission and the reflection spectra obtained?
The authors should describe the integration process of MoS2 on metasurfaces.
More details on biosensing tests should be added, specifically on the type of the target molecule; which target molecule has been investigated? The authors should also compare their reported results with the other reported in literature on the same molecule.
The authors should define the sensitivity of the biosensor; moreover, the authors should add information on the biosensing measurements: experimental conditions, the evaluated concentration range of the target molecule, etc.
The biosensor set up is not clear, please describe.
The manuscript shows lacks in the introduction section, in the material and biosensor set-up, and in the performed experiments.
I reject this manuscript.
Author Response

(The authors gave the same response as above.)

Round 2
Reviewer 1 Report
Good to go.
Reviewer 3 Report
In my opition, after the revision of the manucript lay out, the manuscript can be accepted for publication.